# The Effect of Disulfiram and N-Acetylcysteine, Potential Compensators for Sulfur Disorders, on Lipopolysaccharide-Induced Neuroinflammation Leading to Memory Impairment and the Metabolism of L-Cysteine Disturbance

**DOI:** 10.3390/molecules30030578

**Published:** 2025-01-27

**Authors:** Małgorzata Iciek, Anna Bilska-Wilkosz, Magdalena Górny, Marek Bednarski, Małgorzata Zygmunt, Anthea Miller, Noemi Nicosia, Giorgia Pia Lombardo, Paula Zammit, Magdalena Kotańska

**Affiliations:** 1Chair of Medical Biochemistry, Jagiellonian University Medical College, Kopernika 7, PL 31-034 Cracow, Poland; malgorzata.iciek@uj.edu.pl (M.I.); mbbilska@cyf-kr.edu.pl (A.B.-W.); magdalena.gorny@uj.edu.pl (M.G.); 2Department of Pharmacological Screening, Jagiellonian University Medical College, Medyczna 9, PL 30-688 Cracow, Poland; marek.bednarski@uj.edu.pl (M.B.); malgorzata.zygmunt@uj.edu.pl (M.Z.); anthea.miller@studenti.unime.it (A.M.); paula.zammit.19@um.edu.mt (P.Z.); 3Department of Veterinary Sciences, University of Messina, 98168 Messina, Italy; 4PhD Program in Neuroscience, School of Medicine and Surgery, University of Milano-Bicocca, 20900 Monza, Italy; n.nicosia@campus.unimib.it; 5School of Medicine and Surgery, University of Milano-Bicocca, 20900 Monza, Italy; 6Department of Chemical, Biological, Pharmaceutical and Environmental Sciences, University of Messina, Viale Ferdinando Stagno d’Alcontres 31, 98166 Messina, Italy; giorgia.lombardo1@studenti.unime.it; 7Department of Pharmacy, Faculty of Medicine and Surgery, University of Malta, MSD 2080 Msida, Malta

**Keywords:** disulfiram, hydrogen sulfide, lipopolysaccharide, N-acetylcysteine, neuroinflammation, non-protein thiols, sulfane sulfur, sulfate, cognitive disorders, novel object recognition

## Abstract

Background: The role of sulfur-containing drugs, disulfiram (DSF) and N-acetylcysteine (NAC), in alleviating neuroinflammation is poorly understood. The objective of this study was to examine the effect of DSF and NAC on memory and on the metabolism of L-cysteine and inflammation-related parameters in the cerebral cortex of rats in a model of neuroinflammation induced by the administration of lipopolysaccharide (LPS). Methods: All the treatments were administered intraperitoneally for 10 days (LPS at a dose of 0.5 mg/kg b.w., DSF at a dose of 100 mg/kg b.w, and NAC at a dose of 100 mg/kg b.w.). Behavior was evaluated by the novel object recognition (NOR) test and object location (OL) test, and the level of brain-derived neurotrophic factor (BDNF) was assayed to evaluate neuronal functioning. Cerebral cortex homogenates were tested for hydrogen sulfide (H_2_S), sulfane sulfur, sulfates, non-protein sulfhydryl groups (NPSH), nitric oxide (NO), and reactive oxygen species (ROS) by biochemical analysis. Results: Neither DSF nor NAC alleviated LPS-induced memory disorders estimated by the NOR test and OL test. The studied compounds also did not affect significantly the levels of BDNF, ROS, NO, H_2_S, and sulfane sulfur in the cerebral cortex. However, we observed an increase in sulfate concentration in brain tissues after LPS treatment, while DSF and NAC caused an additional increase in sulfate concentration. On the other hand, our study showed that the administration of DSF or NAC together with LPS significantly enhanced the cortical level of NPSH, of which glutathione is the main component. Conclusions: Our study did not confirm the suggested potential of DSF and NAC to correct memory disorders; however, it corroborated the notion that they reduced oxidative stress induced by LPS by increasing the NPSH level. Additionally, our study showed an increase in sulfate concentration in the brain tissues after LPS treatment, which means the upregulation of sulfite and sulfate production in inflammatory conditions.

## 1. Introduction

Lipopolysaccharide (LPS) is a well-known agent used to induce chronic inflammation. Since it disrupts the blood–brain barrier, it also leads to neuroinflammation, inducing microglial activation [1], and it has been successfully used to study neurodegenerative disorders with memory impairment [2]. Many reports collectively indicate that LPS in mouse models triggers the immune pathway. This causes increased oxido-nitrosative stress and stimulates the release of various proinflammatory mediators [3,4]. Consequently, this leads to the development of behavioral and memory deficits [5].

Previous studies shed light on the role of hydrogen sulfide (H_2_S) in memory formation and storage due to its ability to modulate synaptic plasticity. H_2_S has been recognized as a physiological gasotransmitter next to earlier described nitric oxide (NO) and carbon oxide (CO) [6]. Some research showed cognitive impairment linked with abnormalities in H_2_S signaling in various neurological conditions, including Parkinson’s disease (PD) and Alzheimer’s disease (AD) [7]. H_2_S is regarded as an important neuroprotectant and neuromodulator [8], and it was shown that exogenous administration of H_2_S or its donors reduced LPS-induced memory impairment [9]. Under physiological conditions, H_2_S exists in the free form and also in the form of sulfane sulfur-containing compounds (such as persulfides, polysulfides, and elemental sulfur) [8,10,11]. Based on the study performed in an aqueous solution, it is believed that, at physiological pH and 37 °C, free H_2_S, which is a weak acid, exists mainly as hydrosulfide anion HS^−^ (nearly 80%); about 20% is present as H_2_S, while the sulfide anion S^2−^ is found at very low concentrations [12]. H_2_S is synthesized endogenously from L-cysteine, and its physiological concentration is maintained in a low range due to toxicity at higher concentrations. The bioavailability of H_2_S is regulated through its conversion to sulfane sulfur compounds and by its mitochondrial catabolism yielding thiosulfate and sulfate as the end products [13].

Some promising investigations have shown that the well-known sulfur-containing drug N-acetylcysteine (NAC) may have potential therapeutic effects on memory deficit caused by LPS-induced neuroinflammation [14,15]. Chronic NAC treatment upregulated the expression of synaptophysin (a marker of synaptic plasticity) and caveolin-1 (a major structural protein of caveolae playing an important role in synaptic plasticity, which was downregulated by LPS administration in the medial prefrontal cortex of mice), thereby alleviating the working memory deficit induced by LPS [16]. NAC, a potent antioxidant, was also shown to protect neurons from oxidative damage and to preserve synaptic function by scavenging free radicals and restoring glutathione with a positive effect on the reduction in working memory deficit [17]. Furthermore, existing data suggest that NAC may be effective in reducing cognitive decline in neurological disorders, including AD and PD [18]. NAC is an L-cysteine prodrug and conventionally it acts as a scavenger of reactive oxygen species (ROS), reductant of disulfide bonds, and a precursor for glutathione biosynthesis. Interestingly, a recent study suggested the conversion of NAC to H_2_S and sulfane sulfur species [19].

Another sulfur-containing compound, disulfiram (DSF, also known as tetraethylthiuram disulfide, Antabuse), acts as an inhibitor of aldehyde dehydrogenase (ALDH) and is used in the treatment of alcoholism [20]. DSF is an old drug; however, in recent years many interesting studies have been conducted to discover its biological effects and potential applications in the treatment of various diseases, including cancer [21], parasitic infections [22], HIV, and COVID-19 infections [23,24,25]. DSF has been found to markedly suppress neuroinflammation and dopaminergic neuron loss and to restore motor function. It modulates neuroinflammatory pathways through multiple mechanisms, including the inhibition of inflammatory signaling and enhancement of neuroprotective mechanisms [26,27]. In spite of numerous studies of DSF, the mechanism of its biological activity is still a mystery worth solving. DSF is able to inhibit enzymes, e.g., ALDH, by reacting with important cysteine residues, which could be responsible for its pharmacological activity. Recently, it was reported that DSF acts as an inhibitor of cystathionine β-synthase (CBS), the enzyme responsible for H_2_S synthesis in the brain [28].

In the present study, we examined the effect of DSF and NAC on memory, on the metabolism of L-cysteine (H_2_S, sulfane sulfur, and sulfates), and on inflammation-related parameters in the cerebral cortex in a model of neuroinflammation induced by the intraperitoneal administration of LPS to rats. We wanted to find out whether DSF and NAC can reduce neuroinflammation-induced memory impairment and if this phenomenon is related to their effect on H_2_S synthesis and catabolism. Behavioral tests: a novel object recognition (NOR) test and object location (OL) test were used to assess the working memory of adult Wistar rats, while the level of brain-derived neurotrophic factor (BDNF) was used to evaluate neuronal functioning. Moreover, biochemical assays of H_2_S, sulfane sulfur, sulfates, and non-protein sulfhydryl groups were performed to explain whether H_2_S and sulfane sulfur are involved in the pharmacological action of DSF and/or NAC.

## 2. Results

### 2.1. The Effect of DSF and NAC on LPS-Induced Memory Impairment

There were no statistically significant differences in the locomotor activity (LA) in the open field measured during the first two sessions of behavioral testing between any groups of rats. The LA in each group is shown in Figure 1a,b.

Rats from the inflammation-free control group spent more time investigating the new object in the NOR test than the familiar one. The rats from the group treated with LPS alone had no significant preference in exploration time for the familiar object (A) over the unfamiliar object (B) during the second session of the tests. This group exhibited the lowest discrimination index. The difference between these groups was statistically significant (*p* = 0.0331, Kruskal–Wallis’s test, post hoc Dunn’s test) and the exploration time of the new object was significantly different in these groups. Most animals from the groups administered DSF together with LPS or NAC together with LPS, but also DSF alone or NAC alone, spent more time exploring a new object than a known one. This was evidenced by the positive results of the calculated indices. However, these differences were not statistically significant (Kruskal–Wallis’s test) when compared to both control groups (Figure 1c).

There were no significant differences in the discrimination indices in the OL test among individual groups (Kruskal–Wallis’s test). The results are shown in Figure 1d. However, by analyzing the percentages of animals reacting to objects placed in the open field, it was calculated that, in the control group without induced inflammation, 62.5% of the animals were more interested in a new object (in a new arrangement) while, in the group treated with LPS alone, only 37.5%. In the groups administered DSF together with LPS or NAC together with LPS, only 25% and 33.3% of animals, respectively, were more interested in a new object (in a new arrangement). In contrast, the percentage of rats interested in a new object (in a new arrangement) in groups treated only with DSF or NAC was 62.5% and 71.4%, respectively.

### 2.2. The Effect of DSF and NAC on Body Weight and Food Intake

The mean body weight of rats in the control group without induced inflammation was significantly higher compared to the body weight of rats in the control group with induced inflammation (LPS + vehicle) and in the test groups, LPS + DSF or LPS + NAC, from the 2nd to 10th day of the experiment (two-way ANOVA, F(45, 351) = 9.379, *p* < 0.0001). Figure 2 shows changes in body weight during the experiment. There were no statistically significant differences in body weight in the groups, LPS + DSF or LPS + NAC, compared to the body weight determined in the control group with induced inflammation (LPS + vehicle). From day 7 to 9, in the group treated with LPS + DSF, the body weight of the rats was significantly lower than in the group treated with DSF alone.

The administration of DSF alone or NAC alone had a statistically significant effect on the change in the body weight of the rats. Both of these groups had significantly lower body weight compared to the control group from the fourth or sixth day for NAC or DSF, respectively.

The average food intake by the rats from the control group was statistically significantly higher than the average food intake by rats from the LPS-treated group from the first to every fourth day of administration (two-way ANOVA, F(30,108) = 4.112, *p* < 0.0001). The results are presented in Table 1.

Rats from the groups treated with LPS + DSF, LPS + NAC, or NAC alone consumed significantly less food compared to rats from the control group treated only with vehicles, on all days of administration. Rats from the group treated with DSF alone consumed less food compared to the control group on days 1–4 and on the last day of administration

Rats treated with LPS + DSF consumed statistically significantly less food compared to rats treated with LPS alone on each day of treatment. Rats treated with LPS + NAC consumed significantly less food compared to rats treated with LPS alone on days 2, 3, 5, and 7 of administration. Rats treated with LPS + DSF consumed statistically significantly less food compared to rats treated with DSF alone on each day of treatment.

### 2.3. The Effect of DSF and NAC on BDNF, NO, ROS, NPSH, Sulfates, Sulfides, and Sulfane Sulfur Levels in Cerebral Cortex

The level of BDNF in the cerebral cortex of rats with LPS-induced inflammation was lower when compared to the control group; however, this difference was not statistically significant. In the group treated with LPS + NAC, the level of BDNF was close to the control level, while in the group treated with LPS + DSF, no increase in the BDNF level was observed compared to the LPS alone-treated group. Generally, there were no statistically significant differences in the BDNF levels among all studied groups. The results are shown in Figure 3a.

The level of NO in the cerebral cortex of rats from the group treated with LPS alone did not differ compared with the control group treated only with vehicle. The results obtained for NO estimation are shown in Figure 5B. The medians calculated for the NO level in all tested groups were higher compared to the median calculated in the control group, by over 61%, 48%, 76%, 33%, and 111% for the groups treated with LPS alone, LPS + DSF, LPS + NAC, DSF alone, and NAC alone, respectively. However, only in the NAC alone group was this NO level increase statistically significant.

The ROS level in the group administered LPS alone was statistically significantly higher compared to the level determined in the control group (*p* < 0.0001, Kruskal–Wallis’s test, post hoc Dunn’s test). For the remaining groups, no significant differences were determined compared to the groups treated with saline + vehicle or with LPS alone. The results are shown in Figure 3c.

The level of NPSH determined in the rat cerebral cortex from the LPS-treated group was much lower (by over 80%) compared to the NPSH level in the control group treated only with vehicles (*p* < 0.0001, one-way ANOVA, Tukey’s post hoc test). Figure 4a shows the results. In the groups treated with LPS + DSF or LPS + NAC, the NPSH levels were statistically significantly higher compared to the group treated with LPS alone (one-way ANOVA, Tukey’s post hoc test, F(5, 39) = 62.86, *p* < 0.0001). In the cerebral cortex of rats from the NAC alone-, DSF alone-, and LPS + NAC-treated groups, the levels of NPSH were also higher compared to the level determined in rats from the control group (Figure 4a).

The levels of sulfates determined in the rat cerebral cortex from all LPS-treated groups and from groups treated with DSF alone or NAC alone were statistically significantly higher than sulfates determined in the rat cerebral cortex from the control group (one-way ANOVA, Tukey’s post hoc test, F(5, 35) = 30.31, *p* < 0.0001). The results are shown in Figure 4b. In the rat tissue from the LPS + DSF- or LPS + NAC-treated groups, these levels were also significantly higher compared to the levels determined in rats from the LPS group.

There were no statistically significant differences in the levels of sulfides and sulfane sulfur in the cerebral cortex among all studied groups. The results are shown in Figure 4c,d, respectively.

## 3. Discussion

A ten-day administration of DSF or NAC at doses of 100 mg/kg b.w. in parallel with LPS was unable to protect the animals from spatial and visual memory disorders caused by neuroinflammation associated with LPS administration. However, our study revealed that the administration of DSF or NAC together with LPS could significantly increase the level of non-protein thiols, (mainly glutathione, one of the most important hydrophilic antioxidants), increasing the reduction potential and alleviating oxidative stress induced by LPS administration. Moreover, our study showed an increase in sulfate concentration in brain tissues after LPS treatment, which means the upregulation of sulfite and sulfate production in inflammatory conditions.

In order to evaluate the impact of the test drugs on spatial and visual memory impairment in rats that had previously received LPS treatment, we selected two tests: the NOR test and the OL test. LPS is widely utilized to establish experimental models, including those of memory problems, because the literature data indicate that it effectively produces inflammation in experimental animals [29,30]. The OL test is based on the capacity to find items spatially, whereas the NOR test is based on the ability to visually recognize a novel object. It is normal for rodents—including rats—to be more interested in unfamiliar items in their surroundings than in those they already know. They have no trouble recalling the appearance and placement of objects if their memory is intact. As a result, animals are less interested in items they have already viewed, particularly if they are in the same location [31]. In our study, there were no statistically significant differences among the discrimination indices determined in individual groups in the OL test. The literature also contains reports in which scientists carried out analogous tests (NOR and OL) examining drug effects on memory distorted by the administration of LPS, in which no statistically significant difference was obtained in the discrimination indices calculated for the control group and the group treated with LPS alone in the OL test [32]. Many authors focused their studies (as shown by results in publications) only on the NOR test, without the OL test [33,34]; the reason may be that they did not obtain significant differences in discrimination indices between the control group and the group receiving only LPS. We would like to point out, however, that in our calculations, when the result was zero, it means that the animal did not prefer any of the objects over the other. An index above zero indicates that the animal was more interested in the new object, and, therefore, it was assumed to remember the familiar object and its location in the open field. Unfortunately, the obtained results showed that in rats from all groups administered LPS, i.e., LPS + vehicle, LPS + DSF, and LPS + NAC, most results were below zero, i.e., spatial memory was disturbed. Undoubtedly, the limitation of these studies is the small number of animals used in these experiments. It is known that a larger sample may show more statistically significant results. Due to the principle of limiting the use of animals, we used only 48 animals in our experiment. The results obtained are, therefore, screening data that should be developed using other alternative methods.

Our results indicate that the administration of DSF may influence the reaction of rats in behavioral tests but DSF was not able to clearly reverse memory disorders caused by the administration of LPS, as shown in the NOR test. However, it should be noted that the calculated discrimination indices for the group of rats administered LPS + DSF did not differ statistically significantly from the indices calculated for the group administered DSF alone and from the indices calculated for the group administered vehicles alone. Therefore, the effect of DSF should actually be considered positive. It should be emphasized that such observations were made despite the observed reduction in the intensity of oxidative stress after the use of DSF, as shown by the ROS level in the cerebral cortex (Figure 3c). A lot of studies showed that LPS induces memory impairment via increasing ROS [35,36] and via neuroinflammation [37,38]. Nevertheless, the dose of DSF we used was not able to unambiguously eliminate the disturbances caused by LPS; however, it showed a tendency to decrease the ROS level.

DSF is a hydrophobic, symmetrical molecule. The drug has a strong affinity for protein-bound and -unbound thiols [39]. The literature data indicate that, after intraperitoneal administration to rats, DSF is rapidly absorbed and its concentration in tissues reaches the maximum between 0.5 and 1 h after administration [40,41]. It was found in the liver, heart, kidney, pancreas, thyroid, adrenal, testes, spleen, marrow, and muscle with smaller amounts in the brain and blood [42]. DSF is rapidly metabolized in the liver to water-soluble metabolites, mainly diethyldithiocarbamate (DDC). A total of 93% of DSF metabolites persist in the body for 48 h [40]. These pharmacokinetic properties, its rapid degradation by the glutathione reductase in plasma, and small concentrations achieved in the brain [42] appear to hamper its development as a drug targeted to the brain, but intensive research is being conducted on DSF nanoformulations that achieve very good brain penetration [43]. On the other hand, we believe that a higher dose would cause side effects of an unacceptable level since the dose used in the present study caused decreases in body weight and food intake. In contrast to our study, a recently published paper reported that DSF at a dose of 50 mg/kg b.w. suppressed neuroinflammation and reduced the loss of dopaminergic neurons in a mouse model of PD [26].

We obtained similar results for NAC administration. The literature data in in vitro and in vivo models indicate that NAC has a beneficial effect on alleviating memory disorders in various models [14,15,44,45], especially due to its antioxidant properties, but some in vivo data suggest that NAC is unable to prevent LPS-induced disturbances, e.g., due to the impermeability of the blood–brain barrier (BBB) [1]. NAC is highly hydrophilic (logD − 5.4) and, therefore, has limited capacity to passively cross plasma membranes [46]. However, NAC could be converted to cysteine by aminoacylases in the kidney or other tissues [47], and cysteine is actively transported across the BBB [48]. Undoubtedly, NAC passage through the BBB may increase with disturbances in the functioning and integrity of this barrier [49]. The destruction of the BBB in the model of neuroinflammation induced by LPS administration has been reported, and NAC may be able to inhibit this disturbance [50]. This may be due to the increase in GSH levels in the brain (in astrocytes) after NAC administration [50,51] because cerebral GSH plays an important role in maintaining functional BBB integrity [52]. However, interesting studies conducted by Sakai et al. reported that, although NAC inhibited LPS-induced TNF-α and NO synthesis (in low concentration), its high concentrations (≥30 mM) might cause an increase in mortality of microglia [53].

On the first day of behavioral testing, we telemetrically measured spontaneous activity in an open field, in exactly the same boxes that were used on the following days to determine the effects on memory. On this day, there was no significant difference in animal activity among groups. This is important because it provides a guarantee that the activity of animals did not affect the results obtained in the subsequent tests.

We then determined the effect of administering the tested substances on body weight and food intake. Changes in these parameters, and more specifically lower food consumption and weight loss, clearly indicate that the body reacted negatively to some stimulus, in this case, to inflammation induced by the administration of LPS. These data are consistent with the literature reports [1,54]. In fact, in rats from the LPS-treated group (with inflammation induced), body weight and food intake were significantly lower than in the control group for the first few days. This clearly indicates that the animals felt bad and ate less because of it. At that time, for technical reasons, the impact of the drugs on spontaneous activity was not determined, which would also show whether the animals were sedated. Our results show that, also in the other two groups administered LPS (i.e., LPS + DSF and LPS + NAC), body weight and food intake differed from those parameters determined for the control rats. However, it should be noted that the administration of DSF alone or NAC alone was also associated with a reduction in body weight and food intake. Importantly, food intake in the LPS + DSF-treated group was significantly lower than in the DSF alone group, and rats from this group (LPS + DSF) ate the least and weighed the least throughout the experiment. Undoubtedly, our results confirmed that after DSF the animals consumed less food, which was also previously shown in studies of disulfiram as a potential anti-overeating agent [55], and this may be related to side effects related to the gastrointestinal tract, such as the induction of nausea or dysgeusia [55,56]. Therefore, in subsequent studies we will focus at the cellular level on showing the impact of DSF on the digestive tract—intestines and liver—after the administration of DSF alone or together with LPS.

BDNF is a neurotrophin that is widely expressed as a growth factor in mammalian brains. BDNF plays an important function in neural plasticity, synaptic transmission, and survival [57]. Previous research indicates, for instance, that long-term exercise is connected with an increase in BDNF-positive cells in the hippocampus and cerebral cortex and could improve spatial memory in mice [58]. The upregulation of BDNF by antioxidants had a neuroprotective potential in a rat model of status epilepticus [59], and BDNF upregulation counteracted cognitive deficits induced by ethanol in mice through the enhancement of hippocampal neurogenesis [60]. LPS injection was shown to promote BDNF depletion [61], which may be related with persistent oxido-nitrosative stress and neuroinflammation. Therefore, it was justified to measure BDNF in the rat brains in our experiment. However, we did not obtain significant changes in the levels of this factor among groups.

LPS, several mediators released by microglia, proinflammatory cytokines, and oxidative-nitrosative stress work together to initiate neuroinflammatory processes and neurodegeneration [62]. Our research showed that LPS administration leads to increased oxidative stress (increase in ROS level) and NO production (the mean NO level in the group treated with LPS alone was higher by approximately 60% than the level determined in the control group), which is consistent with the literature data [63,64]. Earlier studies confirmed that even modest increases in the simultaneous production of NO and superoxide could seriously stimulate the formation of peroxynitrite, and even the generation of a low flux of it over long periods of time could result in the substantial oxidation and potential destruction of host cellular constituents, leading to the dysfunction of critical cellular processes [65]. Neuro-oxidative-nitrosative stress may be the molecular basis underlying brain dysfunction including permanent cognitive deficits in neuroinflammation [66]. Our study shows that despite the ability to reduce oxidative stress by increasing the level of NPSH and the tendency to reduce the level of ROS, both DSF and NAC did not have a beneficial effect on the level of NO during neuroinflammation, which may be related to their inability to significantly influence memory impaired by LPS treatment.

Accumulating studies have revealed the protective effects of H_2_S or its donors in neurological diseases. Some recent studies documented also the role of H_2_S in LPS-induced neuroinflammation [9,67,68,69]. H_2_S can regulate cellular functions of target proteins by the persulfidation of cysteine residues, while DSF inhibits enzymes, e.g., ALDH, by reacting with cysteine residues. DSF can react with a protein cysteine residue, forming a mixed disulfide, which is unstable. If, in the vicinity, there is a free cysteine moiety, an intramolecular disulfide bond can be formed [70]. This non-physiological disulfide form of protein can be repaired by the thioredoxin system (Trx); however, if it is inactive due to oxidation (in inflammation), it leads to the accumulation of aggregates and toxic proteins, which should be degraded by the proteasome (Figure 5). Moreover, activity of the proteasome could be inhibited by the complex of DDC with copper [71]. Another possible metabolic pathway of the unstable mixed disulfide formed in the reaction of protein cysteine residue with DSF involves its reduction by GSH that leads to protein glutathionylation (Figure 5). This kind of protein modification is regarded as the protection of protein cysteine residues from oxidation. An increase in protein glutathionylation after the administration of various DSF concentrations in different cell lines was reported previously [72]. Moreover, it has been reported that DSF can inhibit one of the main enzymes responsible for H_2_S synthesis in the brain—CBS [28].

On the other hand, NAC, in addition to its antioxidant properties, can be used as a cysteine precursor for H_2_S and sulfane sulfur production [19]. Therefore, in this study we aimed to check the effect of DSF and NAC on H_2_S metabolism in the cerebral cortex in LPS-induced inflammation. Our results did not reveal the effect of either DSF or NAC on H_2_S and sulfane sulfur levels in the rat cerebral cortex (Figure 4c,d). It means that reactive sulfur species, including H_2_S and sulfane sulfur, are not involved directly in the pharmacological action of DSF and/or NAC in our model. Protein cysteine residues are susceptible to oxidation by ROS, which can lead to irreversible oxidative damage. Reactive sulfur species are able to protect protein cysteine moieties against this damage through their modification by persulfidation or the formation of perthiosulfenic, perthiosulfinic, and perthiosulfonic derivatives (Figure 5). Interestingly, a study by Bora et al. [73] reported that the enhancement of cellular persulfides could mitigate neuroinflammation induced by LPS treatment in the mouse endotoxin shock model. Persulfides, next to polysulfides, are the main species collectively known as the sulfane sulfur-containing compounds, and, due to their antioxidant activity, they potentially can reduce oxidative stress and inflammation. The authors showed that NaHS as an H_2_S donor was unable to alleviate neuroinflammation in a mouse model, while an artificial substrate specific for 3-mercaptopyruvate sulfurtransferase (3-MST), the enzyme that contributes to the synthesis of not only H_2_S but also persulfides, mitigated neuroinflammation in the brain tissue. In light of the above facts and taking into account our results revealing no effect of the test drugs on the cellular sulfane sulfur pool, it seems that none of our tested sulfur-containing compounds can be a substrate for 3-MST for persulfide formation and sulfur trafficking.

In the brain tissues of LPS-treated rats, a significant increase in sulfate concentration was observed (Figure 4b). Sulfate is the end product of mitochondrial H_2_S catabolism and the end product of aerobic L-cysteine metabolism. It was previously demonstrated that in the inflammatory condition, an increase in the level of sulfite was observed in stimulated neutrophils [74]. Moreover, the same study reported a significant increase in serum sulfite concentration after systemic LPS injection to rats. Sulfite can be easily oxidized further to sulfate; therefore, an increase in sulfate concentration observed in our study confirmed the upregulation of sulfite and sulfate production in inflammatory conditions. It was reported that intraperitoneal LPS administration leads to kidney damage, which is associated with increased amounts of serum indoxyl sulfate (one of the uremic toxins). In turn, the chronic administration of indoxyl sulfate causes BBB permeability, leading to neuroinflammation and cognitive impairment. An increased level of indoxyl sulfate in a mice model of chronic kidney disease was reported not only in serum but also in the brain. Behavioral tests conducted on chronic kidney disease mice revealed learning and memory defects [75].

Neither DSF nor NAC, when administered together with LPS, reduced the elevated level of sulfate; on the contrary, after their administration, the sulfate level was statistically significantly higher than in the group treated with LPS alone. Both DSF and NAC are sulfur-containing compounds. DSF is extensively metabolized in the liver, mainly to DDC, which can be then transformed to further metabolites, including carbon disulfide. Thus, the elevated sulfate levels in the DSF-treated group can be the result of the oxidation of carbon disulfide. On the other hand, an increased level of sulfates in the NAC-treated group vs. the control group could be explained as the result of the oxidative metabolism of NAC-derived cysteine. The highest concentrations of sulfates were observed in the LPS + DSF and LPS + NAC groups, which is the result of summing sulfates derived from DSF/NAC metabolism and sulfates formed from sulfites created by LPS action.

Our study showed a drastic decrease in the level of non-protein sulfhydryl groups (mainly glutathione, Figure 4a) in the group treated with LPS alone, indicating a drop in the reducing power and intensification of oxidative processes, which is consistent with an increase in ROS production and sulfate concentration. Both DSF and NAC significantly increased NPSH levels in the brain tissues; therefore, they reduced oxidative stress induced by LPS administration. We also noted that the administration of DSF alone to rats was able to raise the level of NPSH (mainly glutathione) in rat brain tissue. Increases in intracellular glutathione were previously observed in in vitro conditions after the addition of DSF to rat cell cultures [76].

In addition to the small number of animals used in this research, the next limitation of our study is the lack of determination of parameters, such as inflammatory cytokines or C-reactive protein. However, our previous studies, in which we used an analogous LPS dosing regimen in rats and these parameters were examined, clearly showed that neuroinflammation indeed developed after LPS administration to rats in such conditions [54,64]. We also regret that, due to a limited amount of tissue, we were unable to assay individually the concentration of L-cysteine. However, we would like to point out that our study shows preliminary results and indicates their validity, and increasing the number of trials and expanding research could increase the strength of our observations; so, further research in this area is needed.

## 4. Materials and Methods

### 4.1. Chemicals

LPS from Escherichia coli 055:B5 (No. L-2880), Kolliphor EL, disulfiram DSF, N-acetylcysteine (NAC), 2′,7′-dichlorodihydrofluorescein diacetate (DCFH-DA), 2′,7′-dichlorofluorescein (DCF), p-phenylenediamine, zinc acetate, trichloroacetic acid (TCA), gelatin, thionine, Folin–Ciocalteu reagent, 5,5′-dithio-bis-2-nitrobenzoic acid (DTNB), bovine serum albumin (BSA), potassium cyanide (KCN), and potassium rhodanate (KSCN) were provided by Sigma-Aldrich Chemical Company (Darmstadt, Germany). Formaldehyde, ammonia (NH3), barium chloride (BaCl_2_), sodium hydroxide (NaOH), sodium chloride (NaCl), ethanol, ferric chloride (FeCl_3_), hydrochloric acid (HCl), ethylenediaminetetraacetic acid tetrasodium salt (EDTA), sodium dicarbonate (Na_2_CO_3_), potassium sodium tartrate, and copper sulfate (CuSO_4_) were obtained from the Polish Chemical Reagent Company (P.O.Ch., Gliwice, Poland).

### 4.2. Animals

For the tests, 48 male Wistar rats (Rattus norvegicus) were used. They were 5–6 weeks old, with a body weight of approximately 200–250 g. Rats were purchased from the Jagiellonian University Animal House at the Faculty of Pharmacy in Krakow. The animals were housed in standard conditions, with unrestricted access to food and water, in plastic cages that were adjusted to their size. There were two rats in every cage. The temperature in the rooms where the rats were maintained was always 22 ± 2 °C, and there was a 12/12 h light/dark cycle. The experimental procedures were performed from 9:00 a.m. to 2:00 p.m. Handling activities (adaptation time) were used to lessen the stress of the animals during the tests. Following the conclusion of the experiment, rats were given an intraperitoneal dose of sodium thiopental (70 mg/kg body weight) to induce anesthesia. The rats were subsequently decapitated. Next, the cerebral cortex was isolated and cryopreserved at −80 °C.

### 4.3. Experimental Design

Inflammation was induced in rats by injecting LPS at a dose of 0.5 mg/kg body weight (b.w.) for 7 days, as previously described [54]. The study involved 48 rats, divided into six groups: two control groups and four test groups. Each group consisted of eight individuals. The selection of the sample size and group size was based on previous research. Animals were randomly assigned to groups at the start of the experiment. For the first 3 days, the rats were acclimatized; they became accustomed to the environment and researchers. Administration of compounds began on the fourth day and was continued for 10 days. Saline (in group 1) and LPS (in the remaining groups) were administered in the morning (around 9:00). The test compounds (DSF and NAC) and vehicle (in the case of the control groups) were administered after 5 h (around 2:00 p.m.). The doses of DSF and NAC were selected based on previous papers [77,78,79,80]. The test compounds or vehicle were administered intraperitoneally to the following groups:Group 1 (control without inflammation)—saline and vehicle (2.5% Kolliphor EL + 2.5% ethanol in water) in a volume of 0.30 mL/rat;Group 2 (control with inflammation)—LPS at a dose of 0.5 mg/kg b.w., dissolved in saline, and vehicle (2.5% Kolliphor EL for + 2.5% ethanol in water) in a volume of 0.30 mL/rat;Group 3—saline and the DSF dissolved in vehicle (2.5% Kolliphor EL for + 2.5% ethanol in water) at a dose of 100 mg/kg b.w.;Group 4—LPS at a dose of 0.5 mg/kg b.w., dissolved in saline and the DSF dissolved in vehicle (2.5% Kolliphor EL + 2.5% ethanol in water) at a dose of 100 mg/kg b.w.;Group 5—saline and the NAC dissolved in vehicle (2.5% Kolliphor EL for +2.5% ethanol in water) at a dose of 100 mg/kg b.w.;Group 6—LPS at a dose of 0.5 mg/kg b.w., dissolved in saline and the NAC dissolved in vehicle (2.5% Kolliphor EL + 2.5% ethanol in water) at a dose of 100 mg/kg b.w.

On the eleventh, twelfth, and thirteenth days of the experiment, behavioral tests were performed. Throughout the entire experiment, the body weight of rats and food intake were measured using a verified scale (WPT 1C, RADWAG Wagi Elektroniczne, Radom, Poland). The timeline and other details of the experiment are shown in Figure 6.

### 4.4. Behavior Studies

To assess cognitive function, including working memory, the novel object recognition (NOR) test including locomotor activity (LA) and object location (OL) test were performed according to the technique described by Bharani and colleagues [81] with minor modifications [54].

These tests were conducted on three days: on the eleventh, twelfth, and thirteenth days of the experiment. On the eleventh day, rats were placed into an empty box for 5 min twice at 90 min intervals for LA test (two 5 min sessions: TA1 and TA2). This was aimed to acclimatize the animals to the new environment and to minimize the influence of stress factors related to the new environment. LA was measured as described earlier [82]. A tiny radio frequency identification (RFID) transponder (2 mm/5 mm) was previously subcutaneously inserted into each rat in the inguinal fold. This method allowed for stress-free mobility measurement, using TraffiCage (TSE System, Berlin, Germany), to telemetrically determine each animal’s location.

On the twelfth day, the NOR test was performed (two 5 min sessions: TN1, TN2). There was a 90 min pause between each 5 min session. Rats were put into boxes with two identical objects (A, A) during the first session. In the second session, the rodents encountered two distinct objects: a familiar one (object A) and a new one (object B). The time of object exploration was measured for each of the two objects separately.

On the thirteenth day, the OL test was performed (two 5 min sessions: TO1 and TO2). During the first session, the box contained object A, known to the rats from the previous day, and a completely new object, C. After a 90 min break, object C was moved 90° clockwise [54]. The time of object exploration was measured for each of the two objects separately.

Each session was recorded using a camera. The recordings were analyzed by the same researchers, who, during this analysis, were unaware of the treatment the animals received. Appendix A shows photos of the objects used during both tests. These objects were made of plastic, cardboard, and glass, and they were attached to the ground with transparent tape during the experiment. Figure 7 shows the NOR test procedure.

The results obtained in the NOR test and OL test were presented as a discrimination index. This indicator was calculated as follows:$$ID=(T_{new}-T_{known})/(T_{new}+T_{known})$$
where *ID* is discrimination index, *T_new_* is time spent exploring a new object or an object located in a changed place, and *T_known_* is time spent exploring a known object in known place [54].

### 4.5. Biochemical Assays

The frozen cerebral cortex tissues were weighed and immediately homogenized in 0.1 M phosphate buffer, pH 7.4 (1 g of tissue in 10 mL of buffer) for 1 min at 6000 rpm with a homogenizer IKA-ULTRA-TURRAX T10 (IKA Poland sp. z o.o company, Warsaw, Poland). Homogenates were next centrifuged at 2000× *g* for 5 min, and the obtained supernatant was used for biochemical assays.

#### 4.5.1. Brain-Derived Neurotrophic Factor (BDNF) Assay

The concentration of BDNF was determined immunoenzymatically using an ELISA kit (catalog number: E0476Ra, BT LAB, Jiaxing, China).

#### 4.5.2. NO Assay

NO was determined with Griess reaction (Nitric Oxide Assay Kit, catalog number: MAK454, Sigma-Aldrich).

#### 4.5.3. ROS Assay

ROS were assayed according to the method using 2′,7′-dichlorodihydrofluorescein diacetate (DCFH-DA), which was deesterified in brain homogenates to 2′,7′-dichlorohydrofluorescein (DCFH) and then was oxidized by ROS to fluorescent 2′,7′-dichlorofluorescein (DCF) [83]. Briefly, to 40 μL of homogenate, 20 μL of 1.25 M DCFH-DA dissolved in ethanol and 740 μL of 0.1 M phosphate buffer (pH 7.4) were added. The reaction mixture was incubated at 37 °C for 30 min (in the dark). Then, the samples were centrifuged at 9660 × *g* for 8 min at 4 °C. ROS were evaluated using a standard curve for 1 μM DCF. The measurements were conducted using a fluorometer (Aex = 488 nm and Aem = 525 nm).

#### 4.5.4. H_2_S Assay

The level of H2S was determined using a modified method of Shen et al. [84] with fluorometric detection. Aliquots of homogenate (100 μL) were mixed with 100 μL of 1M Tris-HCl buffer, pH 9.5, containing 5 mM EDTA followed by incubation at room temperature (15 min). Next, 200 μL of 1% zinc acetate was added and incubation at room temperature was continued for a further 15 min. Then, 200 μL of p-phenylenediamine (12.5 mM) and 50 μL of 40 mM FeCl_3_ in 6M HCl were added. After incubation for 10 min at a room temperature, the samples were next centrifuged at 12,000× *g* for 10 min, and then fluorescence was measured (Aex = 600 nm and Aem = 623 nm). H_2_S concentrations were read using a standard curve prepared from 10 μM thionine.

#### 4.5.5. Sulfane Sulfur Assay

The level of sulfane sulfur was determined by the method of Wood [85], which is based on the cyanolysis reaction. Briefly, 40 μL of 1 M NH3, 210 μL of distilled water, and 50 μL of 0.5 M KCN were added to 50 μL of homogenate. The samples after mixing thoroughly were incubated at room temperature for 45 min, and 10 μL of 38% formaldehyde solution and 100 μL of the Goldstein reagent containing Fe^3+^ cation were added. The samples were centrifuged at 10,000× *g* for 10 min, and absorbance of the obtained supernatant containing red ferric thiocyanate was measured at a wavelength of λ = 460 nm. The calibration curve was prepared for 1 mM KSCN.

#### 4.5.6. Sulfate Assay

The sulfate level was estimated using the sulfate assay kit according to the manufacturer’s instructions (Sigma-Aldrich, Germany). In this method, inorganic sulfate was precipitated in the reaction with barium sulfate in polyethylene glycol, which stabilized the turbidity. We used gelatin solution instead of polyethylene glycol. Briefly, the homogenate was deproteinized by adding 50% of trichloroacetic acid (TCA) to a final concentration of 2.5% of TCA. The samples were centrifuged at 10,000× *g* for 10 min at 40 °C. Then, 100 µL of the obtained supernatant was mixed with the same volume of BaCl_2_ solution in gelatin, and, after 20 min, turbidity was measured at 450 nm against a blank sample containing 2.5% TCA. The amount of sulfates was calculated based on a standard curve prepared for 5 mM Na_2_SO_4._

#### 4.5.7. Non-Protein –SH Group (NPSH) Assay

Determination of NPSH including glutathione and cysteine was carried out by the Ellman’s reaction [86]. The gist of this reaction is the reduction in 5,5′-dithio-bis2-nitrobenzoic acid (DTNB) by –SH group, yielding a yellow product with maximum of absorbance at 412 nm. Briefly, homogenate was deproteinized by 50% TCA to a final concentration of 2.5% TCA; then, the samples were centrifuged at 10,000× *g* at +4 °C for 10 min. A total of 50 μL of the obtained supernatant was added to the reaction mixture containing 850 μL of 0.2 M phosphate buffer (pH 8.2) and 100 μL of 6 mM DTNB, and absorbance was measured 1 min after addition of the supernatant (λ = 412 nm). The standard curve was prepared with 1 mM glutathione.

#### 4.5.8. Protein Assay

The protein was determined using the Lowry method [87]. Briefly, 100 µL of the sample were incubated with 100 µL of 1N NaOH solution and 1000 µL of copper reagent (2% Na_2_CO_3_, 2% potassium sodium tartrate, 1% CuSO_4_) for 15 min. Then, 100 µL of Folin–Ciocalteu reagent was added and incubated for 30 min at room temperature. Absorbance was read at λ = 500 nm. Bovine serum albumin (BSA) solution was used to prepare a standard curve.

### 4.6. Statistical Analysis

In order to ascertain the statistical significance of the findings, a nonparametric test with Dunn’s post hoc test or one-way or two-way analysis of variance (ANOVA) was employed, with Tukey’s post hoc test. The data are displayed as the mean ± standard deviation (SD) and median. Statistical significance was established for differences when the significance threshold was lower than 0.05 (*p* < 0.05). The Shapiro–Wilk’s test was used for analyzing the normal distribution of the data. For data without a normal distribution, a nonparametric test was used (Kruskal–Wallis’s test); for data with a normal distribution, ANOVA tests were used. The GraphPad Prism 7.0 software (GraphPad Software, La Jolla, CA, USA) was used for statistical calculations.

## 5. Conclusions

The aim of this study was to link the pharmacological effects of DSF and NAC with the metabolism of L-cysteine leading to H_2_S, sulfane sulfur, and sulfates in the cerebral cortex of rats. Our previous study confirmed that LPS caused neuroinflammation [54]; in the present study, we showed that it led to some memory impairments. Our study did not confirm the suggested potential of DSF and NAC to correct memory disorders. The obtained results revealed an increase in sulfate concentration in brain tissues after LPS treatment, which confirmed oxidative stress and inflammatory conditions. In our study, a dramatic decrease in NPSH levels was observed in the cerebral cortex of rats treated with LPS alone and, importantly, both tested compounds, DSF and NAC, were able to restore it to the control level or above. It confirms their potential to increase the reducing power and, in this way, to alleviate oxidative stress induced by LPS administration in brain tissues; but, taking into account all the results obtained in this experiment, it was not sufficient to ameliorate LPS-induced memory impairment.

The obtained results indicated that metabolism of L-cysteine in the brain plays an important role in pathological states leading to memory impairment. Therefore, this study is a prelude to further investigations aimed at better understanding the metabolism of L-cysteine in brain cells and searching for drugs regulating this metabolism in the brain as valuable tools to compensate for sulfur disorders in neuroinflammation.

## Figures and Tables

**Figure 1 molecules-30-00578-f001:**
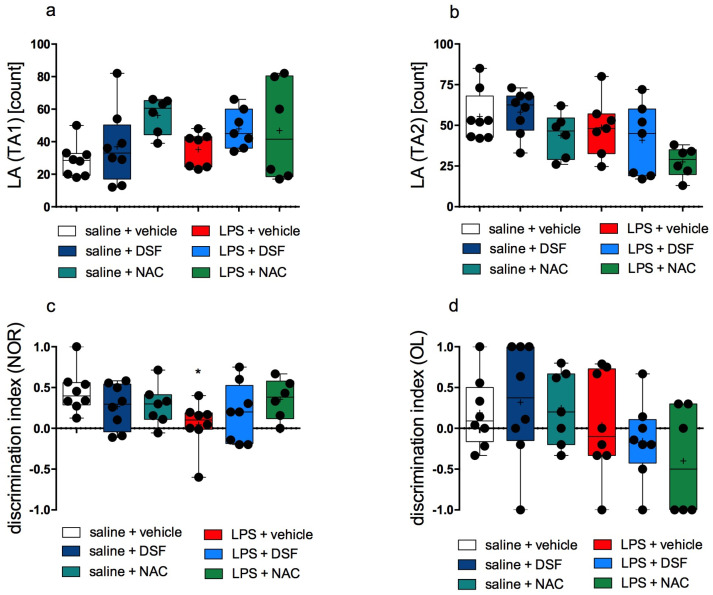
Behavioral tests. (**a**) Locomotor activity (LA) in the first session TA1, (**b**) LA in the second session TA2, (**c**) NOR test–the influence on visual memory (sessions TN1 and TN2), (**d**) OL test—the influence on spatial memory (sessions TO1 and TO2). Median (marked with a line); mean (marked with “+”) ± SD (box); whiskers indicate minimum and maximum values; *n* = 6–8; Kruskal–Wallis’s test; Dunn’s post hoc test; difference vs. control (saline + vehicle) was considered when *p* < 0.05 (“*”).

**Figure 2 molecules-30-00578-f002:**
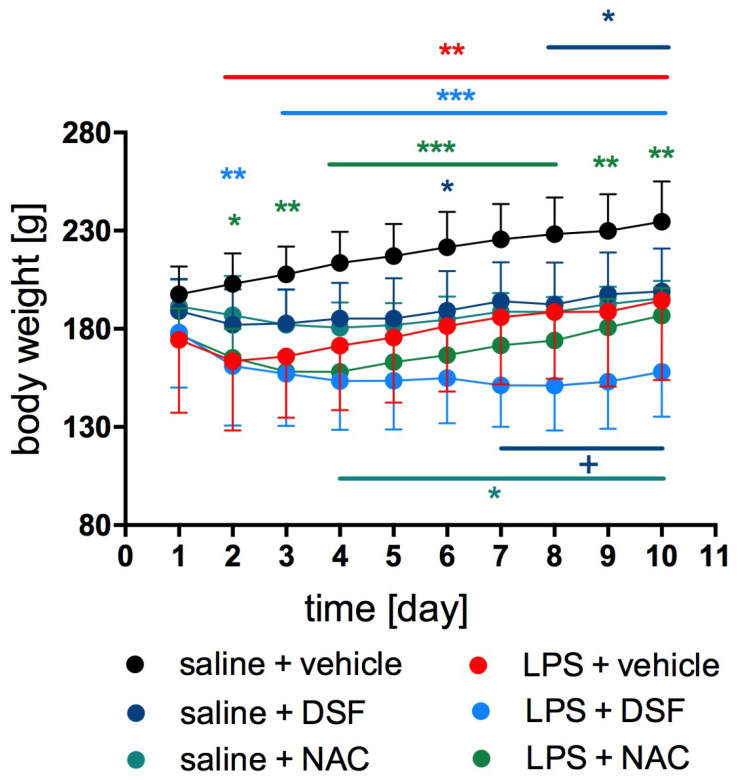
Body weight changes during the experiment in individual groups. Mean ± SD; *n* = 6–8; two-way ANOVA; Tukey post hoc test; difference: “*” (color appropriate to a compared group) vs. saline + vehicle group; “+” LPS + DSF group vs. saline + DSF group; *, + *p* < 0.05; ** *p* < 0.01; *** *p* < 0.001.

**Figure 3 molecules-30-00578-f003:**
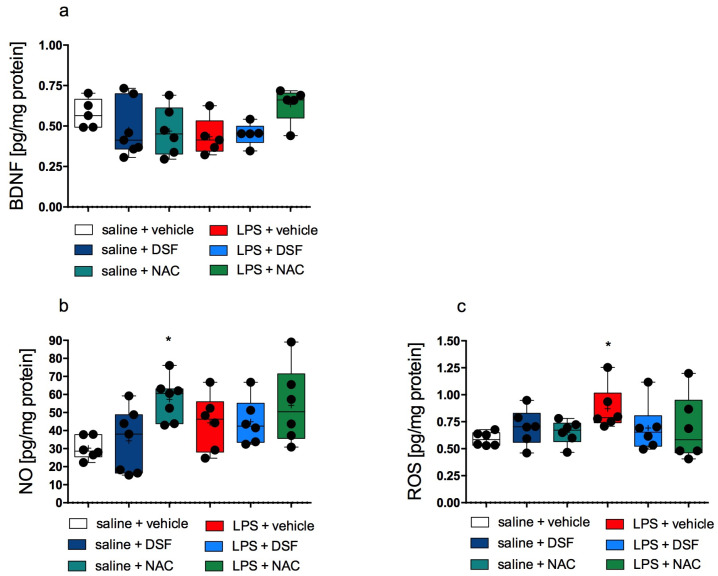
BDNF (**a**), NO (**b**), ROS (**c**) levels in the rat cerebral cortex after treatment. Median (marked with a line); mean (marked with “+”) ± SD (box); whiskers indicate minimum and maximum values; *n* = 5–7; Kruskal–Wallis’s test; Dunn’s post hoc; significant difference vs. control (saline + vehicle) was considered when *p* < 0.05 (“*”).

**Figure 4 molecules-30-00578-f004:**
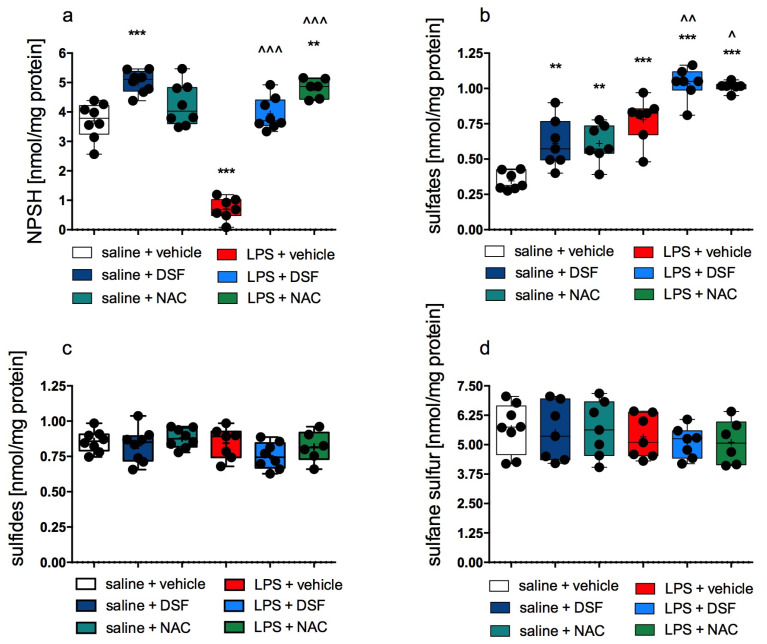
The levels of NPSH (**a**), sulfates (**b**), sulfides (**c**), and sulfane sulfur (**d**) in the rat cerebral cortex after treatment. Median (marked with a line); mean (marked with “+”) ± SD (box); whiskers indicate minimum and maximum values; *n* = 6–8; Kruskal–Wallis’s test; Dunn’s post hoc test; significant difference vs. control (saline + vehicle) is marked with “*”; vs. LPS + vehicle is marked with “^”; ^ *p* < 0.05; **, ^^ *p* < 0.01; ***,^^^ *p* < 0.001.

**Figure 5 molecules-30-00578-f005:**
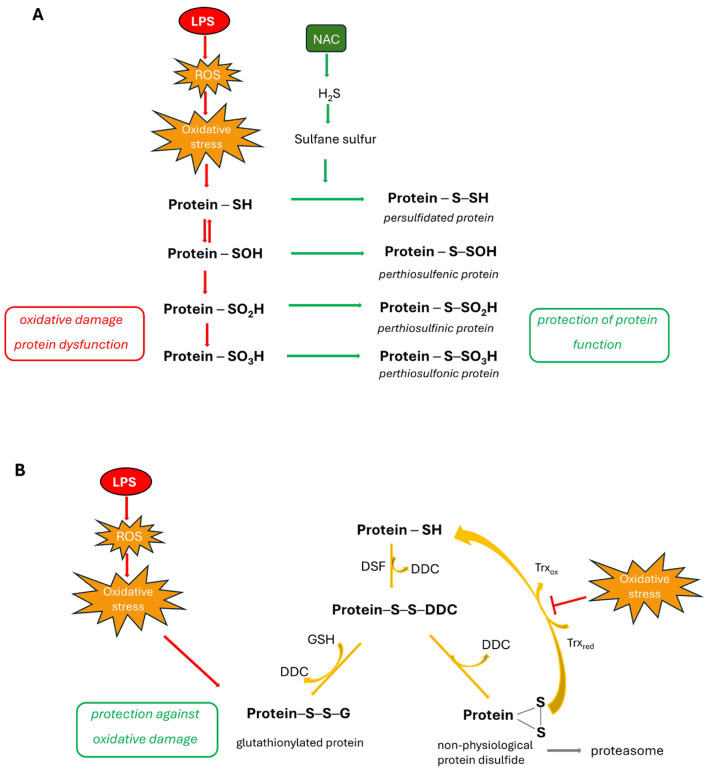
Possible mechanism of NAC (**A**) and DSF (**B**) action in inflammation. (**A**) In the oxidative conditions occurring in inflammation, protein sulfhydryl groups are oxidized to sulfenic acids (-SOH), which is reversible oxidation or to irreversibly oxidized sulfinic (-SO_2_H) and sulfonic (-SO_3_H) acids. NAC, in addition to its direct antioxidant properties, can serve as a precursor of reactive sulfur species (H_2_S and sulfane sulfur) that can modify protein cysteine residues, forming -SSH groups. This process, called persulfidation, is regarded as protein protection against irreversible oxidation and loss of the protein function. Moreover, sulfane sulfur can lead to perthiosulfenic, perthiosulfinic, and perthiosulfonic forms of proteins. The protective effect of this processes is that the modified proteins can be reduced back to native thiols, restoring protein function. (**B**) DSF is a disulfide and can react with protein cysteine residue to form an unstable mixed disulfide, which then, in the thiol-disulfide exchange reaction with reduced glutathione, can form mixed disulfide (protein-SSG). This kind of protein modification (glutathionylation) fulfills protective function against oxidative stress induced by LPS. Another possibility of the mixed disulfide protein-DDC reaction is formation of intramolecular disulfide bond. This takes place when another free cysteine residue is present in the vicinity of this modified by DDC. This non-physiological form of protein can be repaired by the thioredoxin (Trx) system; however, in oxidative conditions, it can be impaired, leading to protein aggregation, ubiquitination, and then degradation by the proteasome.

**Figure 6 molecules-30-00578-f006:**
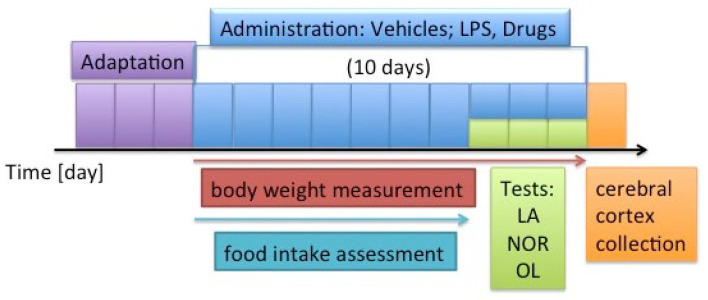
Design of the experiment. The figure shows the time axis [in days], which indicates the durations of the experiments and time points when specific activities in the experiment were performed. Adaptation period—purple (three days), substance administration period—blue (ten days), testing period—green (three days), cerebral cortex collection—orange, body weight measurement—red (ten days), food intake determination—turquoise (seven days). LPS—lipopolysaccharide; NOR—novel object recognition test; OL—object location test; LA—locomotor activity.

**Figure 7 molecules-30-00578-f007:**
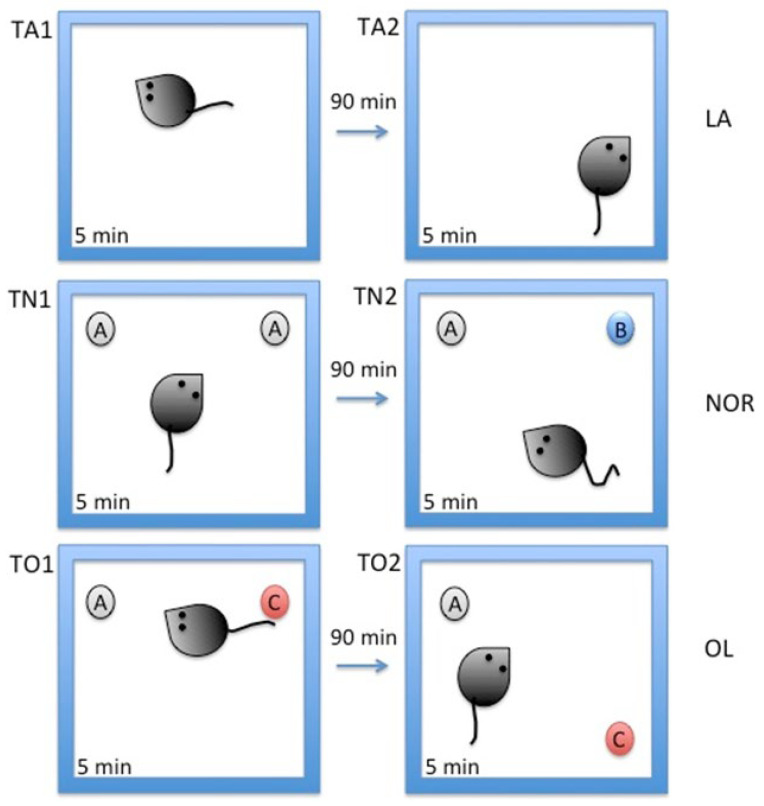
The procedure of the NOR and OL tests. LA test (two sessions: TA1 and TA2); NOR test (two sessions: TN1, TN2); OL test (two sessions: TO1 and TO2); A, B, C—various objects.

**Table 1 molecules-30-00578-t001:** Food intake during the experiment in individual groups.

Day\Group	Saline + Vehicle	LPS + Vehicle	LPS + DSF	LPS + NAC	Saline + DSF	Saline + NAC
	35.63	11.75	2.850	5.250	13.00	21.50
1	±4.230	±6.994	±4.812	±6.775	±4.000	±6.137
		***	***, †	***, ###	***	**
	36.88	16.50	6.300	3.600	21.00	8.650
2	±2.323	±4.123	±6.005	±3.619	±4.546	±1.535
		***	***, ^, †††	***, ^^	***	***
	38.25	25.00	6.750	7.450	28.25	12.60
3	±4.031	±8.124	±2.841	±5.216	±3.403	±3.826
		**	***, ^^^, †††	***, ^^^	*	***, ^
	36.65	23.23	12.95	17.43	21.50	20.53
4	±3.679	±1.652	±6.666	±1.793	±7.122	±3.536
		**	***, ^, †††	***	***	***
	40.28	31.88	15.28	16.25	31.55	21.50
5	±6.058	±1.607	±3.017	±2.500	±3.778	±5.066
			***, ^^^, †††	***, ^^^		***, ^^
	36.58	29.75	10.88	22.25	29.93	26.50
6	±3.443	±4.573	±5.543	±5.252	±2.410	±7.724
			***, ^^^, †††	***		*
	39.00	35.50	16.25	21.50	27.63	18.85
7	±7.071	±9.256	±2.872	±4.041	±3.198	±4.895
			***, ^^^, †	***, ^^	*	***, ^^^

Food intake by two rats (one cage) on successive days of the experiment. Unit: gram; mean ± SD; *n* = 4; two-way ANOVA; Tukey post hoc test; difference: * vs. saline + vehicle group; ^ vs. LPS + vehicle group; † vs. saline + DSF group; # vs. saline + NAC group; *, ^, † *p* < 0.05; **, ^^ *p* < 0.01; ***, ^^^, †††, ### *p* < 0.001.

## Data Availability

The raw data supporting the conclusions of this article will be made available by the authors on request.

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
