# Peer review of "The Effect of Disulfiram and N-Acetylcysteine, Potential Compensators for Sulfur Disorders, on Lipopolysaccharide-Induced Neuroinflammation Leading to Memory Impairment and the Metabolism of L-Cysteine Disturbance"

_molecules, 2025, doi:10.3390/molecules30030578_

Round 1

Reviewer 1 Report

Comments and Suggestions for Authors

Well-designed and well-written research article.

In the Discussion section (lines 439-457) the authors need to briefly include a statement about the possibility of circulating sulfate level being a source of the increased brain sulfate levels that were measured. This could potentially arise from (i) increased circulating sulfate levels due to LPS-mediated damage of the kidneys that reduces filtration of sulfate in the glomerulus; and/or (ii) increased uptake of sulfate from circulation into the brain via LPS-mediated damage of the BBB.

Reviewer 2 Report

Comments and Suggestions for Authors

In this manuscript, authors showed an improved memory of LPS-induced inflammatory Rats using drugs such as Disulfiram or N-acetylcysteine. Authors have utilized several methods such as behavior tests, body weight and food intake, measurement of Nitric oxide (NO), and reactive oxygen species (ROS). To improve the quality of manuscript, I have few suggestions below-

1. In result section, titles for each result should be conclusion or outcomes of the result rather than the name of experiments.

2. The graphs for each results should be thinner to improve the quality. 

3. The food intake data would be clearer to understand if presented in graphs rather than tables.

4. In Figures, write simply as "saline" instead of '0.9 % NaCl'.

2. Line 65, "and 37oC free H2S " 'degree should be in superscript'.

3. Line 87, "DSF, known also as" this sentence should be 'also known as'.

Reviewer 3 Report

Comments and Suggestions for Authors

The manuscript by Kotanska and colleagues provides valuable and timely insights into the role of sulfur-containing drugs in neuroinflammation. This well-written study contributes significantly to the understanding of the mechanisms of DSF and NOV regarding memory, metabolism, and inflammation-related parameters in the cerebral cortex of rats. However, several aspects of the manuscript require improvement for better clarity and scientific rigor.

Minor Suggestions:

  1. Title: The title should be revised to more accurately reflect the study's conclusions rather than its results. Consider highlighting the key findings or implications in the title.
  2. Missing Reference: There is a missing reference for lines 72–74. Please ensure the appropriate citation is included.
  3. Sentence Length: Some sentences in the introduction are overly long and complex. Simplify and break these down to improve readability and flow.
  4. Section 2.1 Behavioral Test: If the results are not statistically significant but are being interpreted as having an effect on memory, further explanation is required. Provide real images from the behavioral tests to support the claims, and clarify whether the tests were conducted in a blinded manner to eliminate potential bias.
  5. Table 1: Add an appropriate and descriptive title for Table 1 to ensure clarity.
  6. Discussion: The discussion would benefit from stronger alignment with previous findings. Reference relevant literature to contextualize the study's results within the broader body of research.
  7. ROS: Include relevant images to support the discussion on reactive oxygen species (ROS) and their role in the observed phenomena.
  8. Conclusion: Rewrite the conclusion to provide a concise and focused summary of the findings, their implications, and potential future research directions.

Reviewer 4 Report

Comments and Suggestions for Authors

The manuscript by Iciek et al. describes the changes induced by DSF or NAC in LPS-induced oxidative stress. They also report a negative result in the correction of memory impairment induced by this compound.

Although the manuscript is well written and organised, some issues should be addressed before it is considered for publication. 

The model shown in Figure 5 isn't based on the results, and to be scientifically sound, the basis of the antioxidant effect of these molecules should be explained in the context of inflammation mediated by LPS. 

Minor issues
-The NOR test decreases significantly with LPS, but it's not significant with LPS+DSF or LPS+NAC (Figure 1). Is this result not a correction of the behavioural impairment by DFS and NAC?
-Figure 2 should be cited in the manuscript (section 2.2).

-NO levels are increased in rats treated with NAC (not mentioned in the results)

Round 2

Reviewer 4 Report

Comments and Suggestions for Authors

The authors have properly resolved all previous issues and doubts.